# Intrauterine Tamponade Balloon for Management of Severe Postpartum Haemorrhage: Does Early Insertion Change the Outcome? A Retrospective Study on Blood Loss

**DOI:** 10.3390/jcm12175439

**Published:** 2023-08-22

**Authors:** Françoise Futcher, Graziella Moufawad, Gabriele Centini, Jad Hayek, Jana Tarchichi, Joseph Bakar, Nassir Habib

**Affiliations:** 1Department of Obstetrics and Gynecology, Moulins-Yzeure Hospital, 10 Av. du Général de Gaulle, 03006 Moulins, France; futcher.f@outlook.fr; 2Department of Obstetrics and Gynecology, Francois Quesnay Hospital, 78201 Mantes-la-Jolie, France; graziella.moufawad@lau.edu (G.M.); jad.hayek01@lau.edu (J.H.); jana.tarchichi@hotmail.com (J.T.); joseph.bakar@ght-yvelinesnord.fr (J.B.); 3Department of Molecular and Reproductive Medicine, University of Siena, 53100 Siena, Italy; centini.gabriele@gmail.com

**Keywords:** intrauterine tamponade balloon, sulprostone, postpartum haemorrhage

## Abstract

The French College of Gynecologists and Obstetricians (CNGOF) recommends the use of intrauterine tamponade balloon (IUTB) in postpartum haemorrhage for bleeding that is refractory after sulprostone before either surgery or interventional radiology. However, the elapsed time between uterotonic drug injection and the insertion of intrauterine tamponade balloon was not reliably assessed. Objective: To evaluate the role of the timing of IUTB insertion and to assess the correlation between the time of insertion and outcome. Methods: A retrospective study in two tertiary care centres, including patients transferred for severe PPH management. Results: A total of 81 patients were included: 52 patients with IUTB inserted before 15 min (group 1) and 29 patients with IUTB inserted after 15 min (group 2). The mean volume of blood loss in the group of patients with IUTB inserted before 15 min was significantly lower than in group of patients with IUTB set after 15 min. Conclusion: An IUTB could be inserted simultaneously with a uterotonic agent, within 15 min and not after 15 min as suggested by local guidelines, but further prospective studies are required to confirm this.

## 1. Introduction

In 2015, over 80,000 deaths worldwide were caused by postpartum haemorrhage (PPH). It is the primary cause of maternal death worldwide [1]. PPH is defined by the Collège national des gynécologues et obstétriciens français as a blood loss of more than 500 mL after vaginal or Caesarean delivery [2]. In October 2017, The American College of Obstetricians and Gynecologists introduced a new classification, where they classified PPH as blood loss of more than 1000 mL or blood loss associated with hypovolaemic symptoms. The main symptoms of hypovolaemia described were palpitations, dizziness, and tachycardia that can shortly be followed by sweating and weakness [3]. However, they considered that an excess of 500 mL after vaginal delivery is an alert where a careful and thorough evaluation should be undertaken. According to this last classification, a blood loss >500 mL must be considered as abnormal [4].

PPH occurs in 1 to 4% of deliveries in economically developed countries and the mortality rate is increasing. In Australia and the UK, haemorrhage features within the top four causes of direct maternal deaths [5]. PPH accounts for 25% of maternal deaths worldwide and occurs within 24 h of delivery in over 50% of cases. Its increase over the years could be explained by an increase in the age at childbirth or rate of caesarean delivery, but the evidence level is low [4,5]. Ford et al. suggested that the increase in PPH may be caused by changes in management, such as an increase in labour induction, more than the risk profiles of women [6]. Bonnet et al. did a retrospective study on the increase in postpartum haemorrhage in Canada. They observed an increase in the number of atonic postpartum haemorrhage. The rate was increased among women who received epidural anaesthesia, induction of labour and ceasarean delivery [7].

The main causes of PPH can be broadly classified as uterine atony, retained placenta, trauma (to the lower genital tract and uterus), and coagulation problems which may be pre-existing or acquired as a result of other pathologies such as disseminated intravascular coagulation [8].

In France, most cases of PPH are caused by uterine atony, which represents 40% of the aetiologies of PPH deaths and 75% of all PPH. Uterine atony is the absence of effective contraction of the myometrium after childbirth and can lead to urgent postpartum haemostatic hysterectomy, which can be considered as a major trauma for women of childbearing age [8].

The higher mortality rate may be explained by a higher incidence of PPH, perhaps due to a high prevalence of PPH risk factors in the French population, or by a higher proportion of severe PPH in France, which could be the result of individual characteristics and/or inadequacies in first-line PPH treatments [1,9,10,11,12].

An intrauterine tamponade balloon [13,14] (IUTB) is a medical device which can be used to treat PPH by avoiding surgery (hysterectomy or compression suture) or uterine artery embolization, particularly in cases of uterine atony. This device has only been studied in a few publications with a low level of evidence [15,16,17].

The IUTB has emerged as a fertility-sparing and life-saving second line treatment option for severe PPH. The Bakri^®^ Balloon (Cook Medical, Bloomington, IN, USA) is currently the most commonly described IUTB device in the literature [18]. Despite the lack of methodological evidence, the IUTB seems very promising: it could reduce the number of surgical operations, which can compromise fertility or lead to postoperative complications [18,19]. No randomized studies have been carried out on the efficacy of IUTB vs. another treatment for PPH. Rosenberg et al. reported an overall success rate of 83.2% of IUTB in PPH management in the largest prospective IUTB cohort [18,20]. Furthermore, another systematic review showed a success rate from 77.5 to 88.8% [21,22]. Few complications have been reported after IUTB insertion, but it does appear to be a risk factor for endometritis [23]. Nevertheless, fertility is preserved [24]. Haemostatic hysterectomy was necessary in only 1% of the women who required a balloon insertion [24].

Recently, Fadel et al. recommended the use of an IUTB in all obstetric departments as first-line management for persistent PPH after medical therapy by sulprostone, being “the least invasive, the easiest to use, and the quickest” with a high efficacy rate [25]. Sulprostone is an analogue of prostaglandin E2 (PGE_2_) which has an oxytocin-like activity [26].

Nevertheless, to date, French guidelines include the possibility of inserting an IUTB but the use of this device is not compulsory [2]. Also, the timing of insertion, volume of inflation, and timing of removal have not yet been standardized.

An IUTB can only be used after the failure of the second-line uterotonic, sulprostone, which is administered at a dose of 500 µg/hour within 20 to 30 min after intravenous oxytocin, and after other bleeding factors have been ruled out. The physician must ensure the absence of retained placenta and carry out a complete examination of the genital tract to exclude trauma before proceeding with an IUTB. The use of an IUTB is an option in the management of PPH. Conventionally, it is inflated with 500 mL of saline or sterile water, and removed after 12 or 24 h of insertion [2,25,26]. However, local protocols may include or may not include the use of an IUTB in the management of PPH, depending on if the teams have the equipment and the training to use them [1]. French guidelines indicate by national professional agreement that the management of PPH must follow protocols established by the establishment [2].

The delay between the injection of a uterotonic drug and the use of an IUTB has not yet been evaluated. Conventionally, it is advised to consider the use of an IUTB after 15 to 25 min, after medication failure as defined by the clinical judgement of the physician [9].

The aim of this study is to define the role of the timing for IUTB insertion and to assess the correlation between the delay of the insertion and the corresponding outcomes.

## 2. Materials and Methods

### 2.1. Study Design

This study was a bicentric retrospective study including all PPH cases from January 2015 to August 2019 in François Quesnay Hospital, Mantes-la-Jolie, and from January 2011 to December 2016 in the tertiary care centre of Beaujon Hospital, Clichy.

Local protocols in both tertiary hospitals for PPH management, irrespective of the type of delivery (caesarean or natural birth), were the use of sulprostone (500 µg per hour over one hour) 20–30 min after intravenous oxytocin with blood loss estimation exceeding 500 mL, and after exclusion of other causes of PPH such as genital tract trauma or retained placenta.

IUTBs have been used since April 2010, and were introduced presumably 15 to 25 min after the failure of the second-line medical treatment. It was inflated with 500 mL of sterile water or saline and deflated 12 to 24 h later. This protocol was the standard algorithm following sulprostone in both units. If bleeding was still active, uterine artery, hypogastric artery embolization, or invasive surgical techniques were indicated.

### 2.2. Inclusion Criteria

All patients with PPH who required an IUTB in the immediate postpartum period after sulprostone failure were included. Blood loss estimation was reported after the placental delivery, at the injection of sulprostone, at the IUTB insertion, and at the end of the management, once the PPH was controlled.

Data collection included demographic, obstetric history and biological factors such as haemoglobin and fibrinogen levels, and the number the blood units and fresh frozen plasma transfused. Data were randomized based on age, gravidity, gestational age at birth, foetal weight, and haemoglobin levels before delivery.

Patients transferred from other hospitals with complete data available were included.

### 2.3. Exclusion Criteria

Patients with incomplete files, with haemostatic surgery or embolization before sulprostone or IUTB, or total blood loss under 500 mL were excluded.

### 2.4. Outcome

The primary outcome was to assess the correlation between the delay in IUTB insertion after medication failure and the total volume of blood loss, and the secondary outcome was to assess the correlation between the delay and the amount of blood loss after IUTB insertion.

Ethics: The local ethics committee was contacted, and approval was given. Data were anonymous, and therefore do not allow patients to be identified.

All transferred patients had an IUTB before transferring to our hospitals. The IUTB was considered efficient if the transferred patient had no intractable bleeding through the cervix or through the balloon drainage channel on arrival at the referral centre. For patients initially admitted and who delivered in the referral centre, the IUTB was considered efficient if the bleeding stopped 15 min after the insertion and inflation of the balloon. At 12 to 24 h after insertion, the IUTB was completely deflated and removed. If the bleeding persisted 15 min after inflation or recurred after deflation, the IUTB was considered inefficient, and emergent arterial embolization or other invasive procedures were immediately performed after the removal of the balloon in the operating room. The procedure was considered a failure if an invasive treatment was necessary, such as uterine arterial embolization, surgery after the IUTB, or surgery after embolization or medical management.

For patients transferred from another hospital, the data regarding the time from the diagnosis to the medication administration and to the decision of IUTB insertion were often lacking before the patients’ admission to the tertiary centre.

### 2.5. Statistical Analysis

Data were analysed using Student’s *t*-test, Fisher’s exact test, and usual statistics (mean, median, standard deviations) with GraphPad Prism version 7.0 for Macintosh (GraphPad Software, San Diego, CA, USA). The data were processed anonymously.

Based on information from other studies, the number of subjects needed for our research using a number of patients out of sight (missing information) of 40%, an alpha risk of 5%, and a power of 80% was 62 patients with 31 patients in each group.

## 3. Results

A total of 133 patients had PPH, between January 2015 and August 2019 at François Quesnay Hospital, Mantes-la-Jolie, and between January 2011 and December 2016 in the tertiary care centre of Beaujon Hospital, Clichy. Only 81 patients were included in the study. The patients were allocated to two different groups, one of 52 patients (62.2%) with IUTB inserted within less than 15 min (group 1) from PPH diagnosis and one of 29 patients (35.8%) with IUTB inserted more than 15 min after the diagnosis (group 2) (Figure 1). The time limit of 15 min was decided according to both local protocols that recommended the insertion of an IUTB at least 15 min and up to 25 min after the injection of sulprostone.

Baseline characteristics for both groups are given in Table 1. There was no statistically significant difference between the two groups regarding demographic and obstetric characteristics. Median age was 32 years old. Overall vaginal delivery was 95%, with 5.7% of caesarean in group 1 and 3.4% in group 2. Gestation and parity were an average of 2.59 and 1.84, respectively, in group 1 and 2.17 and 1.76, respectively, in group 2. Mean foetal weight at birth was 3642 g in group 1 and 3692 g in group 2. Gestational age at birth was 39.52 in the first group and 40.09 in the second. Finally, haemoglobin level at time of PPH diagnosis was significantly different in both groups, with means of 9.07 g/dL (group 1) and 8.08 g/dL (Group 2) (*p* = 0.0274). Moreover, overall aetiology of PPH was mainly uterine atony (88%). In group 1 there were three PPH caused by placenta accreta and one from total retained placenta; in group 2 there were two placenta accreta, two retained placenta and one cervical trauma. Mean delays for IUTB setting in each group were 11.7 min and 29.4 min, respectively. The IUTB was inserted directly (at 0 min) after PPH diagnostic for five patients, and the longest delay before insertion was 60 min after the beginning of PPH for one patient. The number of red blood cells units transfused was significantly different between both groups, 1.61 vs. 3.79, respectively (*p* = 0.00002).

One of the strengths of our study is verifying the assumption that the earlier the IUTB is inserted, the lower the volume of blood loss will be. The overall univariate analysis among the different patient groups revealed that patients who received immediate IUTB insertion had a lower risk of undergoing uterine artery embolization or hysterectomy.

The overall success rate of IUTB use in our study was 91.4%.

In fact, in the group who benefited from early IUTB insertion, there were three intrauterine balloon failures (5.7%); for these three patients, a uterine artery embolization was necessary and sufficient. In group 2, seven failures (24.1%) were reported. Two patients underwent uterine artery embolization, and five patients underwent surgery. Three patients had haemostasis hysterectomy and two patients had uterine compression suture.

We also verified the hypothesis of equality of variance between the two groups with a test of Levene with a significance level of 0.05. Data were analysed using Python version 3.7.5.

For the primary outcome, the average total blood loss was 1150 mL for group 1 and 1627 mL for the second group. We performed a Student’s *t*-test between the two groups to test the difference of mean blood loss. We obtained a *t*-value of −4.19 < 0, giving a *p*-value of 0.00001. The confidence interval was [−702.65; −250.59] with a 0.95 confidence level. Hence, we can conclude that the mean volume of blood loss in the patient group with the IUTB inserted before 15 min was significantly inferior to the group of patients with the IUTB inserted after 15 min.

For the secondary outcome, the average blood loss after IUTB insertion was 256 mL for the first group compared to 545 mL for the second.

We performed a Student’s *t*-test between the two groups to test the difference of mean blood loss volume after the IUTB was used; we obtained a *t*-value of approximatively −3.359 < 0, giving a *p*-value of approximatively 0.0007. The confidence interval was [−459.59; −117.59] with a 0.95 confidence level. These results also show reduced blood loss after the insertion of the IUTB, if it is inserted within 15 min from the beginning of the PPH.

These results confirm our hypothesis that the sooner the IUTB insertion, the more efficient it seems to be. If the IUTB is inserted within 15 min from the beginning of PPH, the total blood loss and the mean blood loss after the insertion up to the end of the episode are significantly lower compared to the group with later insertion.

## 4. Discussion

The intrauterine tamponade balloon is normally used when uterotonics or prostaglandins fail to control PPH. Sterile gauzes were previously used for uterine packing but more recently balloon technology has been used to tamponade postpartum uteri to control haemorrhaging. There are many types of intrauterine balloons [27]. In France, and throughout the world, the Bakri^®^ Balloon seems to be the most frequently used. After insertion, it is recommended to inflate the balloon up to 500 cm^3^ with sterile water or saline. The maximum capacity is 800 cm^3^. The balloon can be filled with a volume less than 500 cm^3^ for smaller uteri but studies have shown that a minimum of 250 cm^3^ is required to achieve haemostasis [12]. In 2009, a report by the Royal College of Obstetricians and Gynaecologists recommended using the Bakri^®^ Balloon in PPH caused by uterine atony [25]. The Bakri^®^ Balloon was further recommended for other causes of PPH in 2011, notably in cases of coagulation disorders or abnormal placental implantation [25]. Yang et al. reported in their study that 52.4% of Bakri^®^ Balloon failures involved placenta accreta but realised that the insertion of a IUTB did help reduce blood loss when a haemostatic hysterectomy was required [28]. The increased use of IUTB has also shown that there are very few complications associated with balloon insertion. Uterine perforation is one of the main complications of IUTB. Two cases have been reported. One was discovered 6 h after insertion in a massive hemoperitoneum in the immediate postpartum period [29]. The balloon was found in the broad ligament. The second case was described following the use of a Bakri^®^ Balloon after removal of a retained placenta, 18 days after childbirth. Due to massive haemorrhage after the placental removal, they inserted a Bakri^®^ Balloon; unfortunately, a laparotomy was required due to persistent haemorrhaging. The intraperitoneal examination revealed that the inflated Bakri^®^ Balloon was located in the left broad ligament. It was not confirmed that the Bakri^®^ Balloon was responsible for the uterine wall dilaceration [30].

Howard and Grobman conducted a study in 2015 on balloon tamponade and morbidity. In this study, 48 patients underwent intrauterine balloon tamponade. They divided the groups into equal quartiles based on blood loss volume from the beginning of the haemorrhage to the balloon insertion [31]. The bottom two quartiles were compared to the upper two quartiles. Results showed that women with IUTB insertion after a lower estimated blood loss volume before insertion had a decreased risk of hysterectomy (0% vs. 38%). This publication further highlights the importance of earlier insertion of an IUTB; however, their main indicator of comparison was blood loss up to balloon insertion, not the time to balloon insertion. Therefore, data extrapolation should be interpreted with caution.

Our study demonstrated the importance of the timing of IUTB insertion by following the rule “the sooner, the better”. Our results show that if the IUTB is placed within 15 min from the beginning of the bleeding, the mean blood loss is significantly decreased compared to a later insertion, i.e., after 15 min. 

The main limitation of this study was that it was a retrospective descriptive study with a limited number of patients. In total, 52 patients were included in the first group, whereas only 29 patients were included in the second group. Another limitation was the caesarean proportion in this study. National maternity units have a caesarean rate around 20% and our study included only 4.9%.

Nevertheless, it is usually known that blood loss is often underestimated. Our study showed that blood loss volume was significantly higher than other studies. The Collège national des gynécologues et obstétriciens français indicates that the systematic use of a collection bag is left to the choice of the teams during delivery; if no bag is used, the estimated blood loss may be lower [2]. A collection bag was used in all deliveries in this study, as indicated by local protocols. Studies have tried to find the most accurate way to estimate blood loss; no better methods were found than visual evaluation and direct estimation by calibrated drape [32]. That is why it is important to look out for hypovolaemia symptoms. Physicians should look out for palpitations, dizziness, and tachycardia that can shortly be followed by sweating and weakness [3].

Earlier IUTB insertion for the treatment of PPH is also supported by other studies. Niola et al. published a study showing a decrease in embolization rate and other surgical procedures, thus decreasing the risks and costs related to these procedures [24].

The efficacy of intrauterine balloon catheters for the treatment for PPH appears similar to other forms of management. In a systematic review, estimated cumulative outcomes showed success rates of 91% for arterial embolization, 84% for balloon tamponade, 92% for uterine compression sutures such as triple uterine ligation, and 85% for iliac artery ligation [33].

Doumouchtsis et al., in 2008, reported a slightly lower efficacy of balloon use; they found a success rate of 80% in the case of refractory bleeding in uterine atony [27]. More recent studies have found a global success rate of IUTB from 88 to 92%; this variation can be caused by the different IUTB devices used [19,34,35,36]. The Bakri^®^ Balloon is the most frequently used, but condom-UTB devices and double-balloon devices have also been described.

The use of IUTB was shown to reduce the need for hysterectomy as a treatment for PPH [21]. Preserving fertility is also an important issue when dealing with PPH; uterine artery embolization and uterine compression suture are the other techniques for uterine preservation.

Kong et al. studied the effect of intrauterine tamponade balloon on fertility outcome. The majority of the patients (87.2%) in the balloon tamponade group had normal menstrual patterns in the 12 months following delivery. After excluding the patients with contraception, the subsequent pregnancy rate was 42.9% (9/21) in the balloon tamponade group compared to 45.9% (28/61) in the control group (*p* = 0.81). Among the nine subsequent pregnancies in the balloon tamponade group, there were two miscarriages, one scar pregnancy, and one induced abortion, while the remaining five were normal pregnancies with full term deliveries without intrauterine growth restriction [37]. Alouini et al. described the absence of impact on subsequent pregnancies. In their study, all patients who expressed a desire for pregnancy became pregnant after an average of 23 months following delivery with PPH and IUTB insertion. Seven patients continued with their pregnancy, one patient had an ectopic pregnancy, and one patient terminated her pregnancy voluntarily [36].

Uterine artery embolization (UAE) is also a way to maintain fertility when treating PPH. Serre-Cousine et al. studied the impact of UAE on fertility. They estimated the pregnancy rate between 50 and 69% [38]. McLucas et al. also studied fertility after UAE; they determined that 48% of women who indicated a desire for children had successful pregnancies with term deliveries of healthy babies [39]. Unfortunately, these results should be used with caution as they cover all causes of uterine artery embolization and not only in the case of PPH.

The other main question concerning the use of an intrauterine balloon is how long should it be left in place. Einerson et al. studied the relationship between clinical outcome and IUTB removal time. In a retrospective cohort, they compared two groups, one with IUTB duration > 12 h and one with IUTB removal 2–12 h after insertion. They found no significant differences in PPH-related outcomes for blood product transfusion, hysterectomy, and admission in intensive care unit. A higher incidence of postpartum fever was associated with IUTB duration > 12 h but hospital stay was not significantly longer for patients [40].

To prevent postpartum endometritis, antibiotics can be introduced. Wong et al. did a retrospective cohort study and demonstrated that prophylactic antibiotics were associated with a reduction of postpartum endometritis. The overall incidence of endometritis was 5% in the group with prophylactic antibiotics vs. 26% in the group without [41]. Franklin-Dumon et al. conducted a retrospective study to compare the incidence of endometritis between patients with PPH, with and without the use of IUTB. All patients received prophylactics antibiotics. They concluded that the use of IUTB was associated with a significantly increased rate of endometritis despite antibiotics [22]. Singh et al. studied the impact of endometriosis complications. They determined that 1 to 4% of patients have complications after acute endometriosis. Infection can lead to necrosis of the uterus but more often we find intrauterine adhesions. These complications affect fertility if not properly treated [42].

An IUTB appears to be an effective option for severe PPH and this study, like many others, shows arguments confirming the need to include it in national protocols.

Finally, the earlier the IUTB is inserted, the less bleeding is experienced, and thus the lower the transfusion rate. As also shown in our study, the transfusion rate was significantly higher in the group where the IUTB was inserted later.

## 5. Conclusions

This study confirms the hypothesis that IUTB inserted before 15 min is associated with significantly less blood loss. As showed by recent studies, IUTB is a novel method and safe treatment to manage severe PPH.

This should be taken into consideration when deciding when to insert an IUTB for women with PPH, especially knowing that postpartum haemorrhage is the leading cause of maternal mortality worldwide.

However, prospective randomized controlled studies will be challenging to design and carry out because of the urgency of such cases and the necessity to make decisions quickly, as well as due to the lack of appropriate resources at many centres.

## Figures and Tables

**Figure 1 jcm-12-05439-f001:**
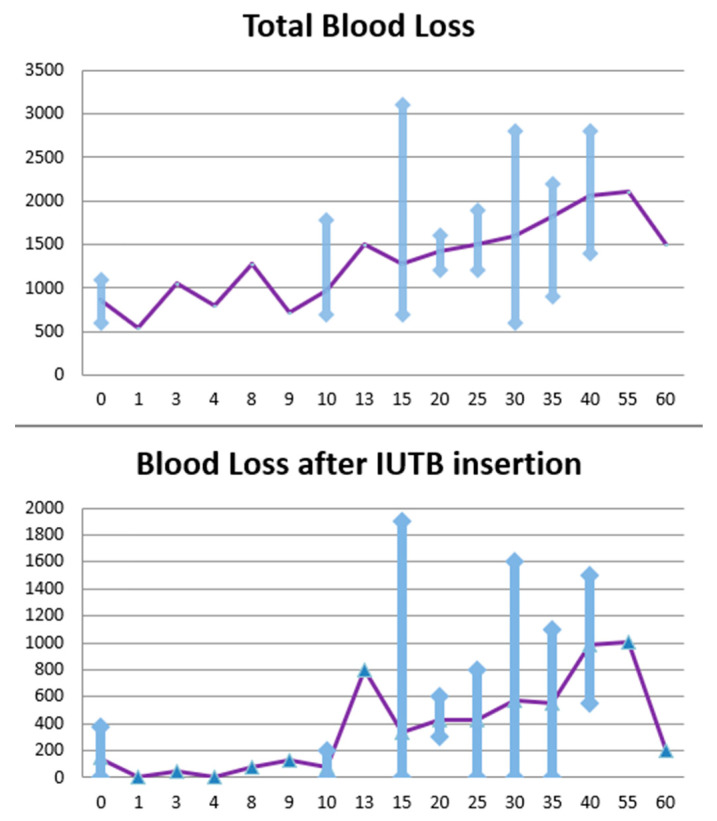
Association between blood loss and IUTB time insertion. The blue triangle are the averages and the purple line is the line to see the evalution of the average. The blue lines veticales lines are the range of values.

**Table 1 jcm-12-05439-t001:** Patient characteristics.

	Group < 15 min (52 Patients)	Group > 15 min (29 Patients)	*p* Value
Age (years)	31.98 ± 5.46	31.01 ± 5.28	0.2176
Gestation	2.60 ± 2.00	2.17 ± 1.31	0.3097
Parity	1.85 ± 1.38	1.76 ± 1.02	0.7658
Gestation age at birth (WG)	39.03 ± 2.05	38.86 ± 3.25	0.7727
Foetal weight (g)	3642.98 ± 586.79	3692.24 ± 752.25	0.7446
Haemoglobin at the time PPH diagnosis (g/dL)	9.33 ± 1.77	8.39 ± 1.86	0.0274
Mean blood loss before IUTB insertion (mL)	899.04 ± 225.89	1087 ± 237.04	0.0007
Total blood loss (mL)	1150.96 ± 485.02	1627.59 ± 498.90	0.00001
Mean blood loss after IUTB insertion (mL)	251.92 ± 344.56	540.52 ± 414.06	0.0012
No. of red blood cell units	1.61 ± 1.79	3.79 ± 3.23	0.00002

## Data Availability

Data available on request due to privacy or ethical restrictions. The data presented in this study are available on request from the corresponding author. The data are not accessible to the public for protection of personal medical data.

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
