# Peer review of "Intrauterine Tamponade Balloon for Management of Severe Postpartum Haemorrhage: Does Early Insertion Change the Outcome? A Retrospective Study on Blood Loss"

_jcm, 2023, doi:10.3390/jcm12175439_

Round 1

Reviewer 1 Report

Abstract/Intro:

p1 l.13: recommends, l23: lower

Why are only two definitions of PPH chosen?

Insufficient explanations of causes for increasing PPH incidence

L62: avoiding surgery? What kind of surgery? What about D&C? Lapatotomy?

l75: There is no guidelines?? What does this mean? Where?

Methods:

What were local protocols? Aren't there French guidelines as mentioned before? What about mode of delivery?

Why were 15 minutes chosen? Insufficient explanation!

IUTB or ITB? Use of abbreviation is inconsistent

AIP? where does this come from all of a sudden?

Results:

Prior blood loss to IUTB insertion?

Table 1: p values?

Prior treatment to IUTB insertion?

l. 224: Lack of accurate blood loss during CS?? What does this mean? It is usually more accurate than in vaginal delivery!

Discussion/conclusion:

How can this be made based on the data provided?

Weakness/strengths insufficiently disussed

The quality of English is quite low! 

Before submitting a paper to any journal,  please make sure that you have someone with a good knowledge of the journal's language to review and edit your manuscript

Author Response

Thank you for reviewing this article. 

I have read and modified my article according to your recommendation and your collegues. 

Reviewer 2 Report

Thank you for giving me the opportunity to review this interesting study; however, I have some comments and recommendations:

Comments:

Title:- Is appropriate for the content of the article.

Abstract:- Represents a suitable summary of the work.

Article content:- The methods and results have been well explained.

Conclusions:- Are justified on the basis of the study.

Recommendations:

Discussion:- Mention the limitations of the study.

References:- Stick to the format of the journal.

General:- Revise language, grammar and syntax.

Revise language, grammar and syntax.

Author Response

(The authors gave the same response as above.)

Reviewer 3 Report

The authors present a manuscript which aims to investigate the efficiency of intrauterine tamponade balloon in the management of severe postpartum hemorrhage. Although the study has been conducted properly and the manuscript has been well written, several corrections should be made to achieve better comprehension. First, the whole manuscript should be edited by a professional in English language. All typographical and grammatical errors should be corrected and there should be consistency in the usage of medical terms (for example, either intrauterine tamponade balloon or Bakri balloon). Second, the authors should rewrite the discussion part so that the authors' findings can be compared with what has been already presented by literature. Moreover, the discussion part should be reorganized and the authors should discuss the limitations and clinical implications of their findings within separate paragraphs. Third, all references that were published before 2008 should be replaced with newer and more up-to-date ones if possible. 

The authors present a manuscript which aims to investigate the efficiency of intrauterine tamponade balloon in the management of severe postpartum hemorrhage. Although the study has been conducted properly and the manuscript has been well written, several corrections should be made to achieve better comprehension. First, the whole manuscript should be edited by a professional in English language. All typographical and grammatical errors should be corrected and there should be consistency in the usage of medical terms (for example, either intrauterine tamponade balloon or Bakri balloon). Second, the authors should rewrite the discussion part so that the authors' findings can be compared with what has been already presented by literature. Moreover, the discussion part should be reorganized and the authors should discuss the limitations and clinical implications of their findings within separate paragraphs. Third, all references that were published before 2008 should be replaced with newer and more up-to-date ones if possible. I recommend that the manuscript can be accepted for publication in Journal of Clinical Medicine after necesssary corrections have been made.

Author Response

(The authors gave the same response as above.)

Reviewer 4 Report

This study was designed to assess the correlation between the delay in IUTB insertion and total blood loss. However, there are several queries and suggestions;

1. Please reconsider the title of the study, which should reflect the methodology and the objective or hypothesis of this study.

2. Please state how to calculate the sample size of this study.

3. Why did the authors use the 15-min cut off time to separate the patients into 2 group? Each group should have the equal number of patients.

4. The correlation should be tested in statistics and the results should be shown. Please revise the figures illustrating the correlation.

5. Was the blood loss estimated accurately? This is an important issue as the estimated blood loss was the main outcome of this study.

6. Please reconsider the methodology and then rewrite the results and the discussion.

7. The conclusion should be more concise.

Author Response

(The authors gave the same response as above.)

Round 2

Reviewer 4 Report

The sample size should be calculated for the results to be reliable. Please reconsider this important aspect. The results could not be interpreted in the inadequate sample size.

Please check and correct spelling.

Author Response

Thank you for your answer.

we have added to sample size calculation. they have been added in the statistical analyses section.

Spenling has been cheaked.

Thank you
